# Predicting the Initial Conditions of the Universe using a Deterministic Neural Network

**Vaibhav Jindal[1,2,*]**
vaibhav.jndl@gmail.com

**Albert Liang[1,3,*]**
ajliang@cs.cmu.edu

**Aarti Singh[1]**
aartisingh@cmu.edu

**Shirley Ho[4,5,6]**
shirleyho@flatironinstitute.org

**Drew Jamieson[7]**
jamieson@mpa-garching.mpg.de

## Abstract

Finding the initial conditions that led to the current state of the universe is challenging because it involves searching over an intractable input space of initial conditions, along with modeling their evolution via tools such as N-body simulations which are computationally expensive. Recently, deep learning has emerged as a surrogate for N-body simulations by directly learning the mapping between the linear input of an N-body simulation and the final nonlinear output from the simulation, significantly accelerating the forward modeling. However, this still does not reduce the search space for initial conditions. In this work, we pioneer the use of a deterministic convolutional neural network for learning the reverse mapping and show that it accurately recovers the initial linear displacement field over a wide range of scales ($< $ 1-2% error up to nearly $k \simeq 0.8 - 0.9 \ \mathrm{Mpc}^{-1} h$), despite the one-to-many mapping of the inverse problem (due to the divergent backward trajectories at smaller scales). Specifically, we train a V-Net architecture, which outputs the linear displacement of an N-body simulation, given the nonlinear displacement at redshift $z = 0$ and the cosmological parameters. The results of our method suggest that a simple deterministic neural network is sufficient for accurately approximating the initial linear states, potentially obviating the need for the more complex and computationally demanding backward modeling methods that were recently proposed.

## 1 Introduction

The evolution of our universe can be uniquely determined by its initial conditions and the laws of physics. To understand this cosmic history, astrophysicists use a large number of surveys (Amendola et al., 2018; Spergel et al., 2015) and simulations (Bagla, 2002; Villaescusa-Navarro et al., 2020).

---

[1]Machine Learning Department, Carnegie Mellon University
[2]Cohesity
[3]Two Sigma Investments
[4]Center for Computational Astrophysics, Flatiron Institute
[5]Department of Physics & Center for Data Science, New York University
[6]Department of Astrophysical Sciences, Princeton University
[7]Max Planck Institute for Astrophysics
[*]Work done while at Carnegie Mellon University

NeurIPS 2023 AI for Science Workshop.

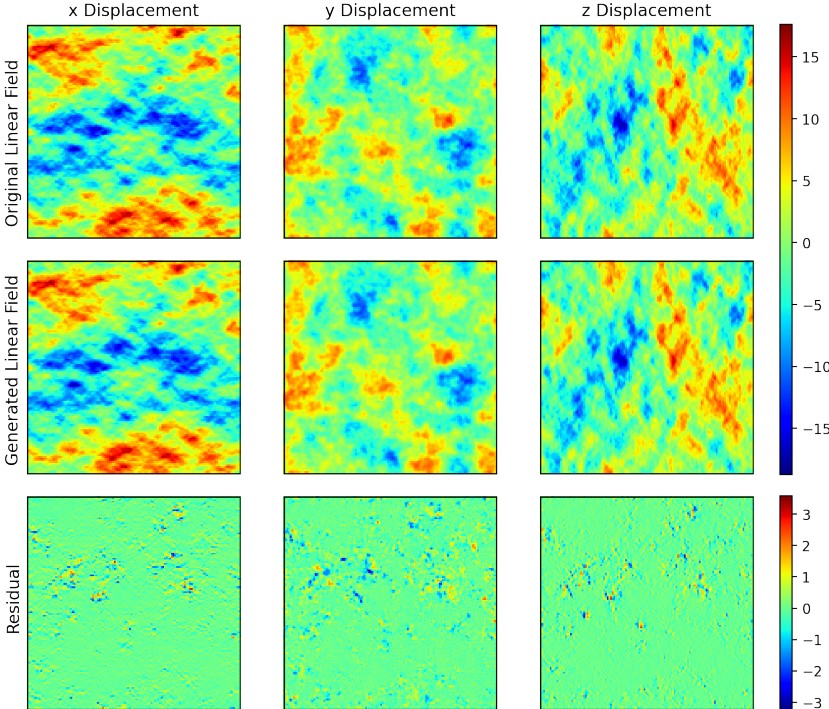

Figure 1: Qualitative comparison of the $x, y$, and $z$ displacements for a $128 \times 128$ slice of particles from a linear field sampled using the training parameters and the corresponding linear field predicted by our inverse model.

These simulations compute the gravitational evolution of a system of N-body particles given a set of nearly uniform and typically Gaussian initial conditions, representing the early universe. These simulations, though, come with a substantial computational cost and demand an extensive investment of both time and computational resources.

In recent years, deep-learning techniques have been shown to be extremely helpful in accelerating the forward modeling process (He et al., 2019; Jamieson et al., 2022) by several orders of magnitude. These deep-learning models learn the mapping between pairs of inputs and outputs from numerical N-body simulations and act as fast and accurate approximators for these simulators.

The problem of inferring the initial state that generates a specific redshift $z = 0$ state of the universe is an inverse problem and poses a more difficult challenge. Inverse problems are hard as they require a search over an intractable space of all possible input configurations, and typically involve a one-to-many mapping if learned as a reverse mapping. Standard neural networks are one-to-one mappings and are thus not expected to work well in these problems. Sampling approaches such as Hamiltonian Monte Carlo used by Bayesian Origin Reconstruction of Galaxies (BORG) (Jasche and Wandelt, 2013; Jasche and Lavaux, 2019) are computationally prohibitive, requiring thousands of CPU hours to generate a single posterior sample. One could resort to more complex generative neural networks such as the recently proposed diffusion models (Legin et al., 2023). However, they are significantly more difficult and expensive to train compared to deterministic neural networks. Moreover, as we show in this paper, they are potentially an overkill to this problem as the distribution of initial conditions is typically Gaussian, where a simple deterministic neural network like ours already works surprisingly well.

Importantly, cosmological simulations are often deterministic, and therefore they are reversible in principle. The one-to-many backward problem arises primarily due to numerical and computational errors which get exacerbated only at small scales which are dominated by nonlinear effects, causing divergent backward trajectories. This motivates the approach we demonstrate here: training a standard

deterministic neural network to learn the reverse map and output the initial states of the N-body simulations using the final states as input.

We show that despite the one-to-many nature of the reverse mapping at small scales, a simple deterministic neural network can do a surprisingly excellent job of predicting the initial states not only at large scales but even down to relatively small scales ($k > 0.1 \text{ Mpc}^{-1} h$) where the nonlinear dynamics of gravitational clustering become important. In fact, our model continues to have $< 1 - 2\%$ error down to $k \simeq 0.8 - 0.9 \text{ Mpc}^{-1} h$. Our results empirically motivate the use of neural networks as approximate inverse-mapping black boxes that could directly generate reliable initial states for a given output state, which could then be used as starting points for more fine-grained sampling-based inverse modeling methods.

## 2 Background

Consider an N-body system with particles distributed on a uniform grid with positions $\mathbf{q}$. Let $\mathbf{\Psi}_{ZA}(\mathbf{q})$ be their linear Zel'dovich approximation (ZA) approximation at redshift $z = 0$ (current time). Thus, the final positions of the particles when they evolve linearly is

$$\mathbf{x}_{lin}(\mathbf{q}) = \mathbf{q} + \mathbf{\Psi}_{ZA}(\mathbf{q}). \tag{1}$$

Let the final nonlinear displacement of the particle initially at grid site $\mathbf{q}$ be $\mathbf{\Psi}_{NL}(\mathbf{q})$. Thus, the final positions of the particles at redshift $z = 0$ under nonlinear evolution is

$$\mathbf{x}_{nonlin}(\mathbf{q}) = \mathbf{q} + \mathbf{\Psi}_{NL}(\mathbf{q}). \tag{2}$$

In this work, we investigate the problem of predicting the linear displacement field at redshift $z = 0$, given the non-linear displacement field and the cosmological parameters.

## 3 Methodology

We train a convolutional neural network (CNN) that takes the nonlinear displacement field $\mathbf{\Psi}_{NL}(\mathbf{q})$ at redshift $z = 0$ and the value of $\Omega_m$ as inputs and predicts the linear displacement field $\mathbf{\Psi}_{ZA}(\mathbf{q})$ at redshift $z = 0$. We directly use the V-Net architecture used in (Jamieson et al., 2022) and train it using 100 pairs of nonlinear and linear displacement fields. In terms of the training procedure, our method is almost identical to (Jamieson et al., 2022), with the main difference being that we pass the nonlinear displacement field as the input to our model and ask it to predict the linear displacement field. One of the key advantages of using a simple deterministic V-Net, instead of a generative model such as a diffusion model, is the reduced computational complexity during both training and inference phases. Deterministic networks do not involve complex stochastic sampling processes, eliminating the need for numerous forward passes which makes them significantly faster and more resource-efficient than the generative models.

For our experiments, we train our model using simulations of $128^3$ uniformly distributed particles in a square box with a side-length of $250 \text{ Mpc}/h$. This corresponds to a Nyquist wavenumber of $k = 1.608 \text{ Mpc}^{-1} h$, the theoretical limit beyond which we can't trust the predictions of any model for this setting. For the training data, we randomly generated 100 linear fields for a fixed set of cosmological parameters ($\Omega_m = 0.300, \Omega_b = 0.050, h = 0.700, n_s = 0.965, \sigma_8 = 0.799$). Since running N-body simulations on these initial conditions are too computationally expensive, we use the emulator by (Jamieson et al., 2022) to generate their corresponding nonlinear displacement fields as our model's training targets, which are already highly accurate for our purpose.

The architecture of our model involves three downsampling and upsampling components each and provides a field-of-view of $96^3$. That is, a focal particle's initial displacement is predicted based on its environment out to its $48^{\text{th}}$ neighbors on the initial particle grid, corresponding to a distance of $93.75 \text{ Mpc}/h$. This finite field-of-view has a benefit: the evolution of the system on large scales is accurately described by linear theory, so the V-Net model preserves this linear evolution on scales larger than the field-of-view while allowing for nonlinear evolution at smaller (medium range of) scales.

On very small scales, the particles clustered tightly into dark matter halos where their orbits become complicated and difficult to predict. This small-scale clustering imposes a resolution limit on the

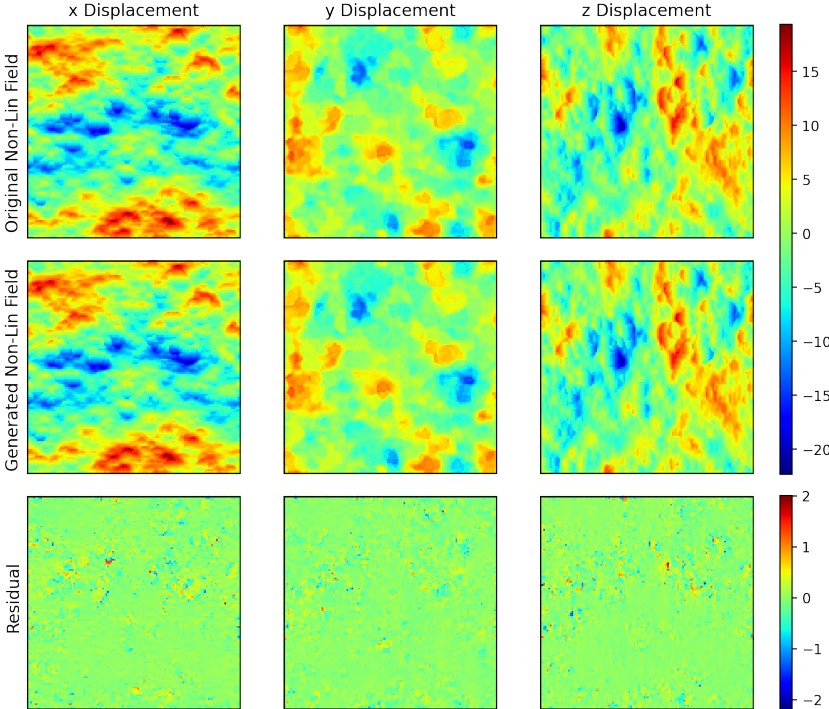

Figure 2: Qualitative comparison of the $x, y,$ and $z$ displacements for a $128 \times 128$ slice of particles from the original nonlinear field and the nonlinear field generated from the linear field predicted by our inverse model. The nonlinear field is generated from the forward modeling emulator (Jamieson et al., 2022). The slice shown here is the same as the one shown in Figure 1 for a one-to-one comparison.

inverse mapping, making it a one-to-many mapping as numerical errors and floating point precision make the neural network model unable to fully encode the detailed textures of this virialized motion. This limitation is also present in the forward model (Jamieson et al., 2022) and the diffusion model (Legin et al., 2023), which tends to make halos more diffuse on small scales than in the simulations, blurring the details of these sharp structures. For the inverse mapping, the initial linear displacements are blurred for particles that end up inside of halos, limiting the accuracy of the inverse model predominantly on these very small scales as shown in Section 4.3.

Sections 4.1, 4.2, and 4.3 show the in-distribution performance of our model on unseen pairs of fields generated from the forward direction emulator (Jamieson et al., 2022) with the same cosmological parameters as its training data. We also evaluate the out-of-distribution (OOD) performance of our model on the Quijote simulation suite (Villaescusa-Navarro et al., 2020) in Section 4.4.

## 4    Results and Analysis

We evaluate our method's performance both qualitatively and quantitatively with a variety of analyses and statistics.

### 4.1    Qualitative Analysis

Figure 1 shows the $x, y,$ and $z$ direction displacements for a $128 \times 128$ slice of the linear displacement field and the corresponding linear field predicted by our inverse model. The original linear field was sampled using the cosmological parameters of the training data. Qualitatively, the predictions of our model match very well with the original linear field that we wanted to predict.

For further evaluation, we used the forward-direction emulator again to generate the nonlinear field when fed the predicted linear field and compared it with the original nonlinear field in Figure 2. We find these slices match very closely with insignificant residuals.

## 4.2 One-Point Statistics

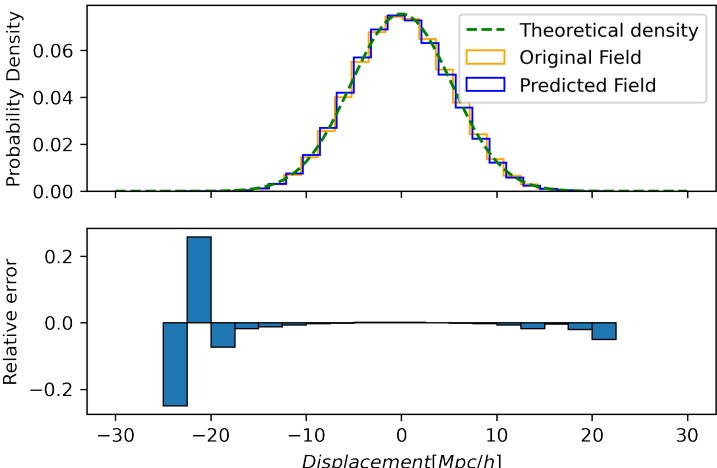

Figure 3: Distribution of the displacements of particles for a given linear field and the corresponding linear field predicted by our model. The distribution has been calculated by considering the $x, y$, and $z$ displacements of $128^3$ particles. Relative error denotes the relative errors in probability density for different bins.

The initial conditions of the simulation are set up in Fourier space, with each mode of the displacement field drawn from a random Gaussian distribution with a variance determined by the linear power spectrum $P(k)$, which is determined by the cosmological parameters. This construction yields a coordinate-space displacement field that is also Gaussian, so its statistics are uniquely determined by the two-point correlation function, which is simply the Fourier transform of the power spectrum. To demonstrate that we have accurately recovered the initial conditions, we must show that the output of our model has both the correct power spectrum and that its statistical distribution is Gaussian.

We plot the histogram of the probability density of the displacements of the particles for the original linear field and the one predicted by our model (Figure 3). We expect the distribution of displacements to match $\mathcal{N}(0, 5.275)$ for the simulation setting with $128^3$ particles in a box size of 250 Mpc/$h$. Specifically, the variance of displacements along the $i^{\text{th}}$ Cartesian direction is given by:

$$\sigma_i^2 = \int \frac{\mathrm{d}k^3}{(2\pi)^3} \frac{(k^i)^2}{k^4} P(k), \tag{3}$$

where the integral is over all wave vectors in Fourier space, $k \equiv |\vec{k}|$ is the magnitude of the wave vector and $k^i$ is its $i^{\text{th}}$ component. We numerically evaluate this integral on a Fourier space grid with the same dimensions as the initial simulation grid.

From Figure 3 we see that the statistics of our model output agree well with the expected Gaussian distribution. The bin counts also match the particular realization from this distribution of the target data out to displacements of $\sim 10$ Mpc/$h$, or $2\sigma_i$. Extremely large displacements indicate a particle either flowing outwards in a rare, extremely underdense environment or inwards towards a rare, high-density peak. Based on the residuals, we see that the tails of the distribution predicted by the model are somewhat smaller than the target data, indicating the model misplaces these particles. This is unsurprising due both to the rarity of these trajectories and to the fact that these particles are the most affected by extreme non-linearities in their environments, which exacerbates the one-to-many problem.

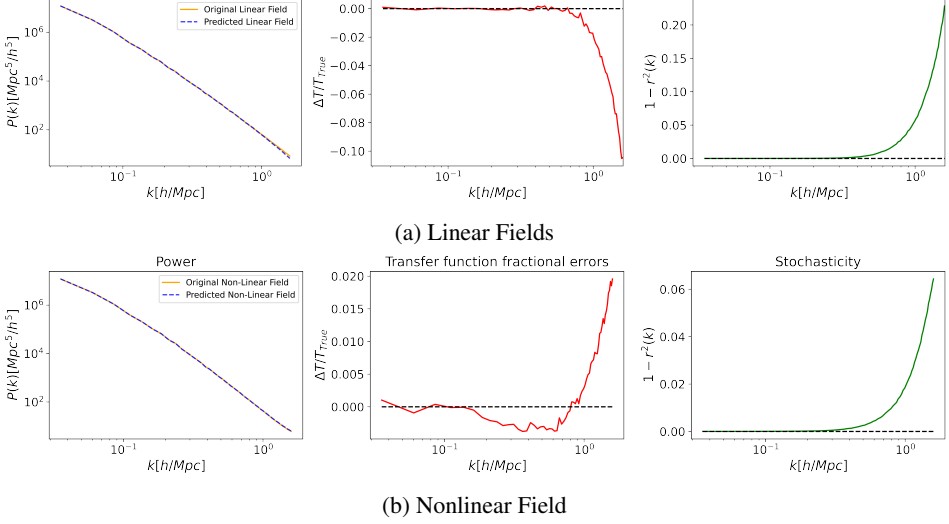

(a) Linear Fields

(b) Nonlinear Field

Figure 4: Two-point correlation comparison between (a) the original linear field and the linear field predicted by our model, and, (b) the original nonlinear field and the nonlinear field generated from the predicted linear field given by the inverse model.

### 4.3 Two-Point Correlation Comparison

The displacement power spectrum for a displacement field $\mathbf{\Psi}$ for wavenumber $k$ is defined as

$$P(k) = \sum_{i \in \{x,y,z\}} \langle \mathbf{\Psi}_i(k) \mathbf{\Psi}_i(k) \rangle. \tag{4}$$

Using this, we define the transfer function as

$$T(k) = \sqrt{P(k)}, \tag{5}$$

and the correlation coefficient as

$$r(k) = \frac{P_{pred \times true}(k)}{\sqrt{P_{pred}(k) P_{true}(k)}}, \tag{6}$$

where $P_{pred}(k)$ is the displacement power spectrum of the predicted field, $P_{true}(k)$ is the ground truth power spectrum, and $P_{pred \times true}(k)$ is the cross power spectrum between the predicted and the ground truth fields. Furthermore, we now define the transfer function fractional error,

$$\frac{\Delta T(k)}{T(k)} = \sqrt{\frac{P_{pred}(k)}{P_{true}(k)}} - 1, \tag{7}$$

to measure the discrepancy between amplitudes of the predicted and the true fields. We also define stochasticity,

$$1 - r^2(k) = 1 - \frac{P^2_{pred \times true}(k)}{P_{pred}(k) P_{true}(k)}, \tag{8}$$

to capture the excess fraction of correlation in the prediction of our model that cannot be accounted for in the target data. For an ideal match between the target and the predicted field, the values of both these quantities should be exactly zero. Figure 4a shows the performance of our model in terms of these quantities. We see that from large scales down to scales where $k \simeq 0.8 - 0.9 \ \mathrm{Mpc}^{-1} \ h$, the model achieves 1-2% percent-level accuracies for both the transfer functions and stochasticities. Note that non-linearities become important at scales where $k > 0.1 \ \mathrm{Mpc}^{-1} \ h$, so the model is able to accurately learn the inverse mapping even in the moderately nonlinear regime for in-distribution fields.

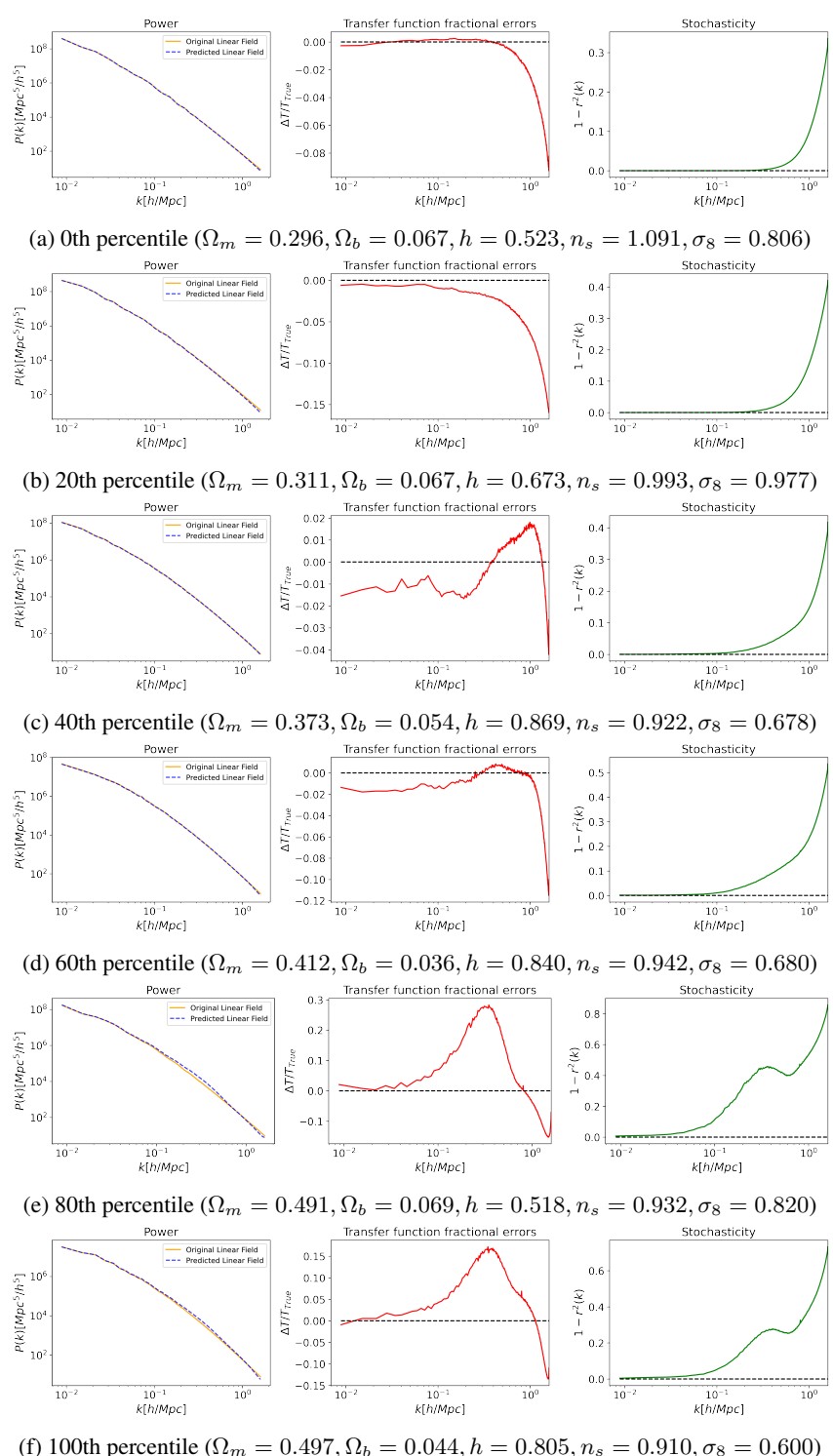

(a) 0th percentile ($\Omega_m = 0.296, \Omega_b = 0.067, h = 0.523, n_s = 1.091, \sigma_8 = 0.806$)

(b) 20th percentile ($\Omega_m = 0.311, \Omega_b = 0.067, h = 0.673, n_s = 0.993, \sigma_8 = 0.977$)

(c) 40th percentile ($\Omega_m = 0.373, \Omega_b = 0.054, h = 0.869, n_s = 0.922, \sigma_8 = 0.678$)

(d) 60th percentile ($\Omega_m = 0.412, \Omega_b = 0.036, h = 0.840, n_s = 0.942, \sigma_8 = 0.680$)

(e) 80th percentile ($\Omega_m = 0.491, \Omega_b = 0.069, h = 0.518, n_s = 0.932, \sigma_8 = 0.820$)

(f) 100th percentile ($\Omega_m = 0.497, \Omega_b = 0.044, h = 0.805, n_s = 0.910, \sigma_8 = 0.600$)

Figure 5: Power spectra comparison between the linear fields and the inverse model predictions for the 0th, 20th, 40th, 60th, 80th, and 100th percentile OOD Quijote simulation data points.

To further test the reliability of our model, we generated the nonlinear displacement field from our predicted linear field by using the forward direction emulator. Since the mapping from the nonlinear displacement field to the linear displacement field is one-to-many, it becomes critical to see that this field matches the actual nonlinear field. Figure 4b shows the power spectra comparison between

these two nonlinear displacement fields. We again see a good match for large to medium scales ($k < 0.5 \ \mathrm{Mpc}^{-1} \ h$).

## 4.4 Out-Of-Distribution Evaluation

Given that our model is trained on simulations generated by a neural network emulator (Jamieson et al., 2022) with fixed cosmological parameters, there is a legitimate concern about its ability to generalize to actual N-body simulations with different sets of cosmological parameters. To assess this out-of-distribution (OOD) performance, we conduct a series of experiments on simulations from the Quijote suite (Villaescusa-Navarro et al., 2020).

The Quijote suite provides 2000 linear and nonlinear fields, each consisting of $512^3$ particles that are uniformly distributed in a cube of side length $1000 \ \mathrm{Mpc}/h$. To quantify the dissimilarity between these simulations and our training data, we rank them based on the sum of their relative percentage differences in $\Omega_m$ and $\sigma_8$ from our training values, since these two parameters have the most significant impact on the simulated distributions. This dissimilarity metric is then used to evaluate the model's performance on different percentiles, with the 100th percentile simulation being the most OOD. Notably, our model's performance is unaffected by changes in the box size, as long as the mean particle density remains constant.

To evaluate the model's performance on OOD Quijote simulation data, we qualitatively compare the $x, y$, and $z$ displacements of a $512 \times 512$ slice from the linear and nonlinear fields, similar to Section 4.1. The slices for the 0th percentile and the 20th percentile simulations are presented in Figures 8 and 11 respectively. Additionally, the one-point statistics for these simulations are provided in Figures 6 and 9, and the two-point correlation comparisons are presented in Figures 7 and 10 respectively.

To examine how the model's performance decays as the nonlinear input gradually gets more OOD, we plot the two-point correlations for a range of percentiles in Figure 5 and Figure 12. As expected, for both the predicted linear field and the regenerated nonlinear field, the model performs excellently for the 0th and 20th percentiles but gradually gets worse for higher percentiles.

## 5 Conclusion

In this work, we pioneer using a CNN to predict the initial linear field of an N-body simulation given the final nonlinear field and the cosmological parameters. We show that despite the many-to-one nature of the inverse mapping, our CNN can still recover the linear fields for a wide range of scales, even including the smaller scales where the nonlinear physics of gravitational clustering becomes important. In addition, we empirically demonstrate that our model generalizes reasonably well to OOD cosmological parameters. This suggests that neural networks can be used as approximate, inverse-mapping black boxes for generating trustworthy initial states for more fine-grained sampling-based inverse modeling methods. Overall, our work demonstrates that despite being much simpler and computationally cheaper than other approaches such as BORG and diffusion models, a deterministic neural network is already capable of reconstructing surprisingly accurate initial conditions of the universe.

In future works, we plan to explore sampling-based methods (e.g. Hamiltonian Monte Carlo (Radford, 2012), Active Learning (Ren et al., 2021), etc.) with initial states proposed by our model. We expect that these methods will refine the initial conditions, achieving high accuracy at even smaller scales. Another direction for future work is to enhance our model to additionally output uncertainty estimates. This will involve uncertainty quantification (UQ) methods such as optimizing the Gaussian negative log-likelihood loss (Nix and Weigend, 1994), re-architecting our model as a Bayesian neural network (Goan and Fookes, 2020), or augmenting our model with conformal predictions (Angelopoulos and Bates, 2022).

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

# A Appendix

## A.1 Results for the 0th percentile OOD setting of cosmological parameters
$(\Omega_m = 0.296, \Omega_b = 0.067, h = 0.523, n_s = 1.091, \sigma_8 = 0.806)$

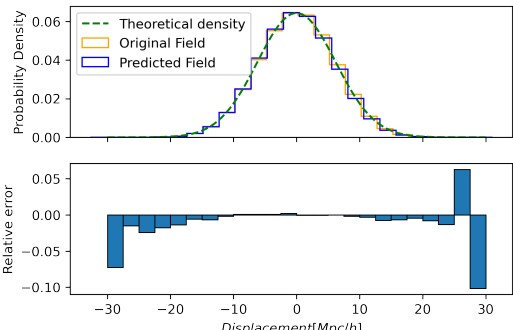

Figure 6: Distribution of the displacements of particles for the 0th percentile OOD Quijote simulation linear field and the corresponding linear field predicted by our inverse model. The distribution has been calculated by considering the $x, y$, and $z$ displacements of $512^3$ particles separately. Relative error denotes the relative errors in probability density for different bins. The theoretical distributions for this setting is $\mathcal{N}(0, 6.193)$ (Equation 3).

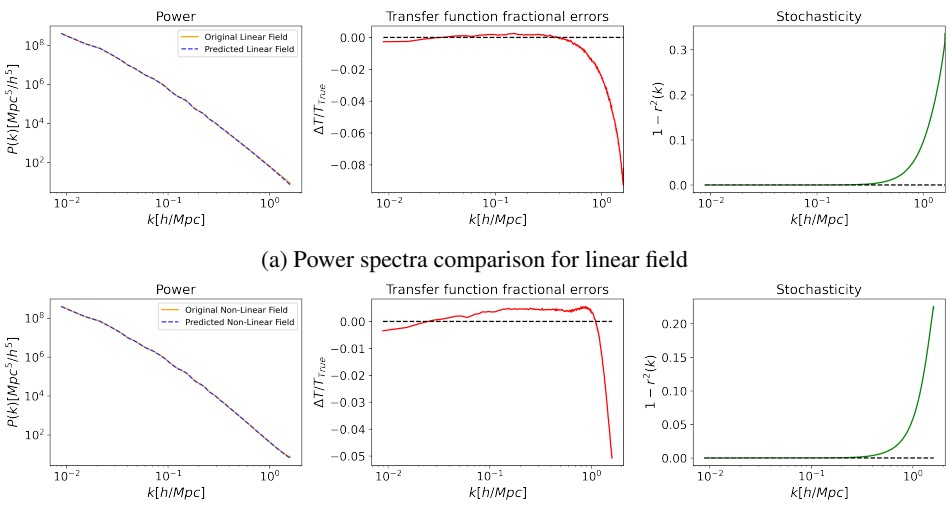

(a) Power spectra comparison for linear field

(b) Power spectra comparison for nonlinear field

Figure 7: Power spectra comparison for the 0th percentile OOD Quijote simulation data point. (a) Comparison between the original linear field and the linear field predicted by our inverse model. (b) Comparison between the original nonlinear field and the nonlinear field predicted by the forward emulator (Jamieson et al., 2022) for the linear field predicted by our inverse model.

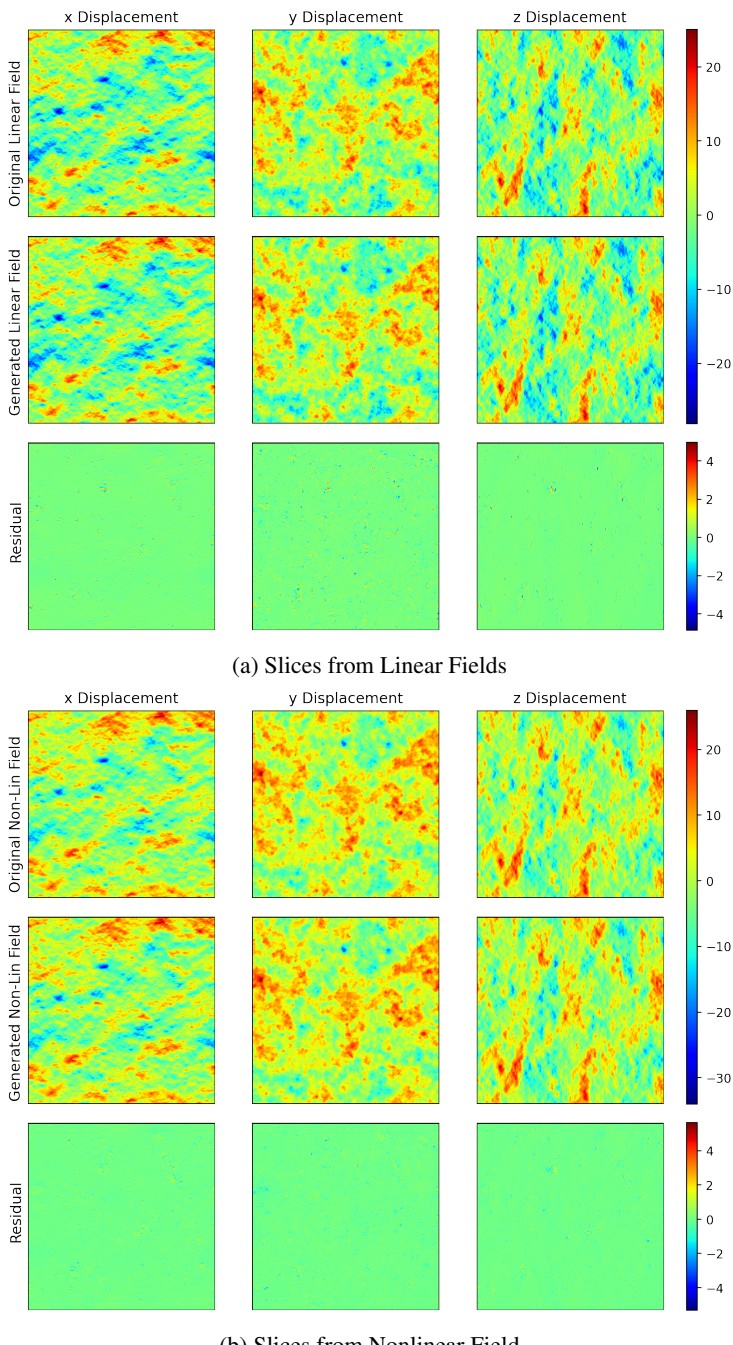

(a) Slices from Linear Fields

(b) Slices from Nonlinear Field

Figure 8: Qualitative comparison of the $x, y,$ and $z$ displacements for a $512 \times 512$ slice of particles for the 0th percentile OOD Quijote simulation data point. (a) Slice from the original linear field and the corresponding linear field predicted by our inverse model. (b) Slice from the original nonlinear field and the nonlinear field predicted by the forward emulator (Jamieson et al., 2022) for the linear field predicted by our inverse model.

## A.2 Results for the 20th percentile OOD simulation from the Quijote suite
$(\Omega_m = 0.311, \Omega_b = 0.067, h = 0.673, n_s = 0.993, \sigma_8 = 0.977)$

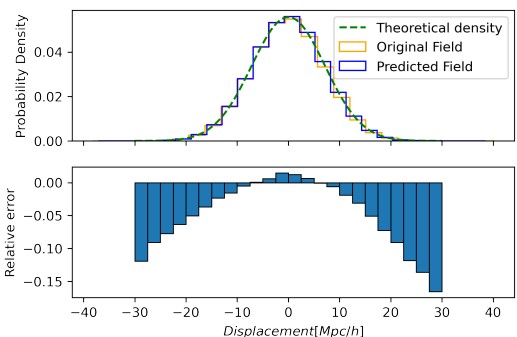

Figure 9: Distribution of the displacements of particles for the 20th percentile OOD Quijote simulation linear field and the corresponding linear field predicted by our inverse model. The distribution has been calculated by considering the $x, y,$ and $z$ displacements of $512^3$ particles separately. Relative error denotes the relative errors in probability density for different bins. The theoretical distributions for this setting is $\mathcal{N}(0, 7.159)$ (Equation 3).

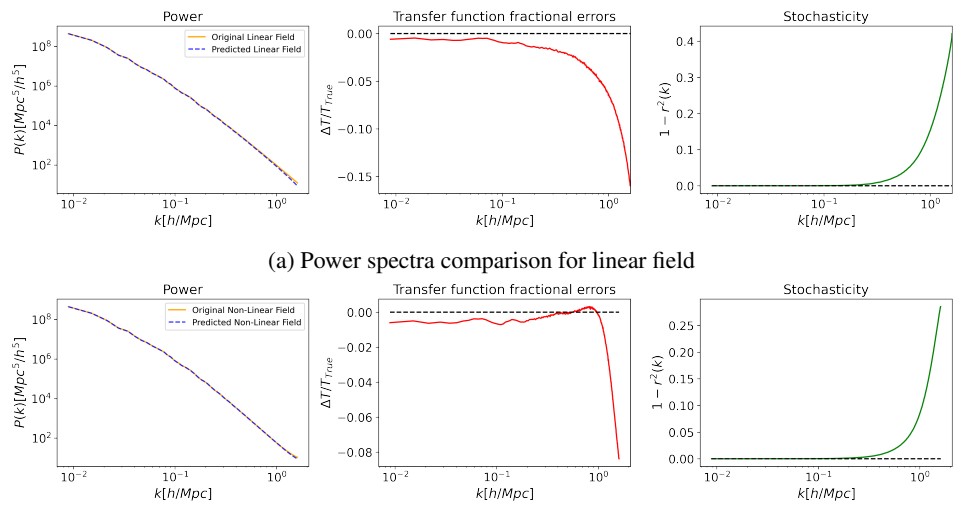

(a) Power spectra comparison for linear field

(b) Power spectra comparison for nonlinear field

Figure 10: Power spectra comparison for the 20th percentile OOD Quijote simulation data point. (a) Comparison between the original linear field and the linear field predicted by our inverse model. (b) Comparison between the original nonlinear field and the nonlinear field predicted by the forward emulator (Jamieson et al., 2022) for the linear field predicted by our inverse model.

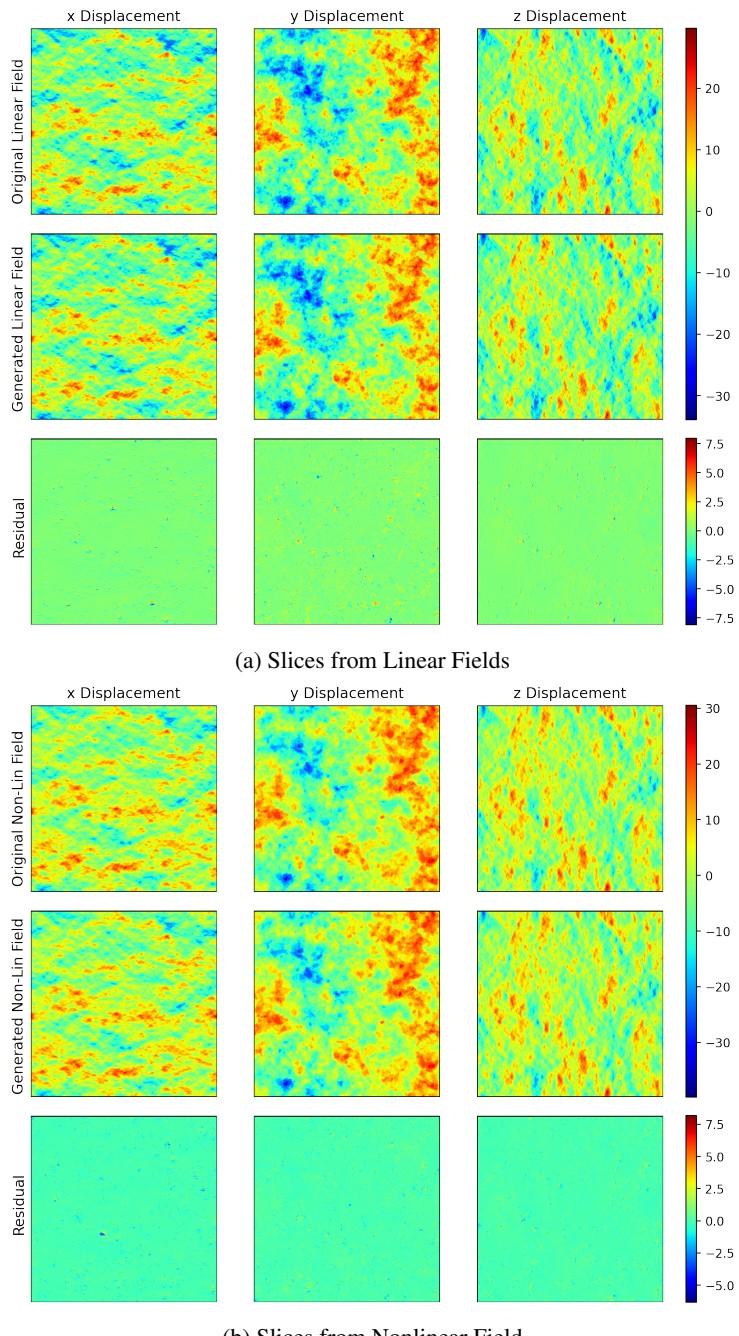

(a) Slices from Linear Fields

(b) Slices from Nonlinear Field

Figure 11: Qualitative comparison of the $x$, $y$, and $z$ displacements for a $512 \times 512$ slice of particles for the 20th percentile OOD Quijote simulation data point. (a) Slice from the original linear field and the corresponding linear field predicted by our inverse model. (b) Slice from the original nonlinear field and the nonlinear field predicted by the forward emulator (Jamieson et al., 2022) for the linear field predicted by our inverse model.

### A.3 Power Spectra Analysis for Nonlinear Fields in OOD settings

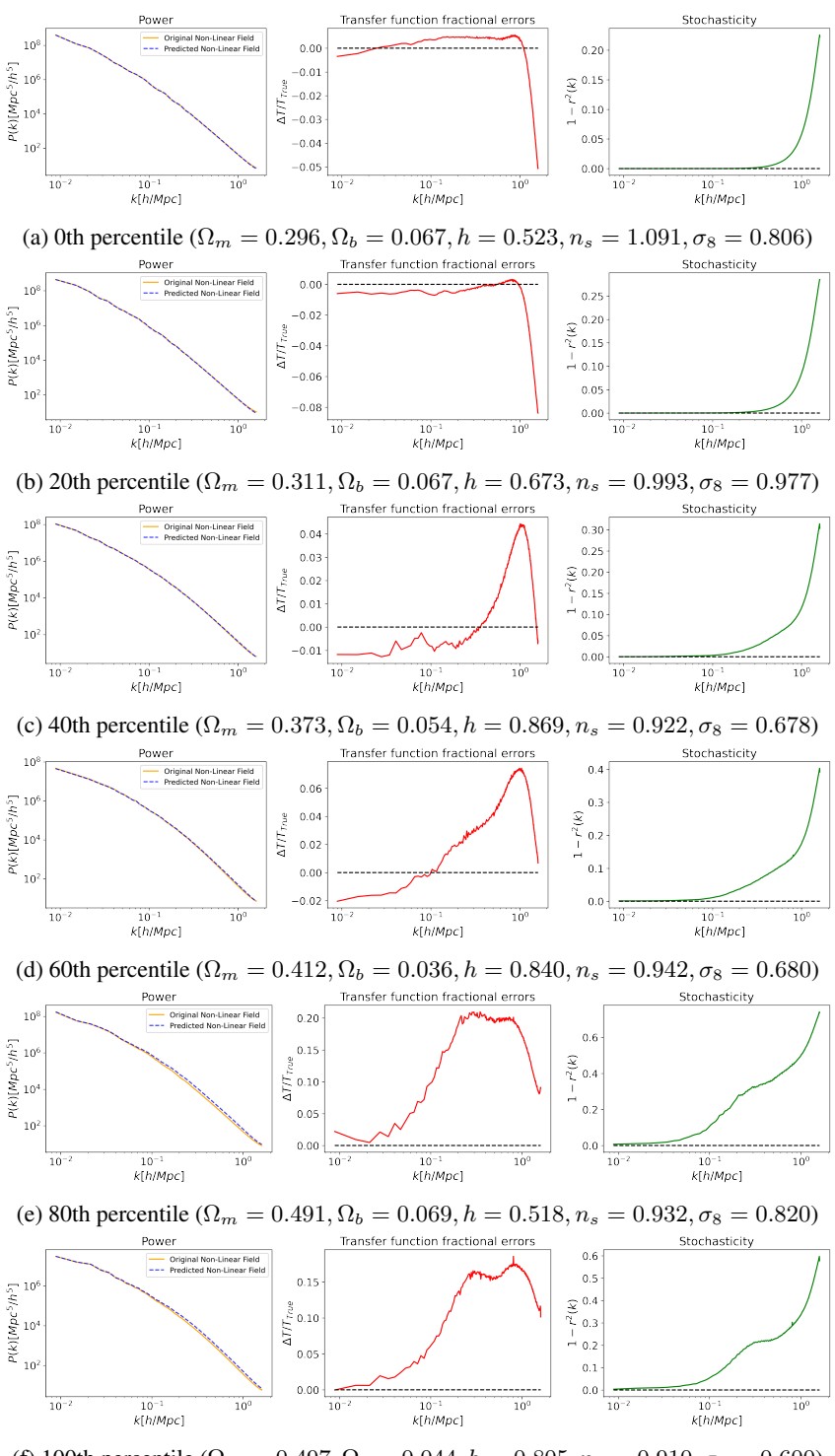

(a) 0th percentile ($\Omega_m = 0.296, \Omega_b = 0.067, h = 0.523, n_s = 1.091, \sigma_8 = 0.806$)

(b) 20th percentile ($\Omega_m = 0.311, \Omega_b = 0.067, h = 0.673, n_s = 0.993, \sigma_8 = 0.977$)

(c) 40th percentile ($\Omega_m = 0.373, \Omega_b = 0.054, h = 0.869, n_s = 0.922, \sigma_8 = 0.678$)

(d) 60th percentile ($\Omega_m = 0.412, \Omega_b = 0.036, h = 0.840, n_s = 0.942, \sigma_8 = 0.680$)

(e) 80th percentile ($\Omega_m = 0.491, \Omega_b = 0.069, h = 0.518, n_s = 0.932, \sigma_8 = 0.820$)

(f) 100th percentile ($\Omega_m = 0.497, \Omega_b = 0.044, h = 0.805, n_s = 0.910, \sigma_8 = 0.600$)

Figure 12: Power spectra comparison between the actual nonlinear field and the nonlinear field generated by the forward emulator for the linear field predicted by our inverse model for the 0th, 20th, 40th, 60th, 80th, and 100th percentile OOD Quijote simulation data points.

