# OpenReview forum: "Predicting the Initial Conditions of the Universe using a Deterministic Neural Network"
_NeurIPS.cc/2023/Workshop/AI4Science — NeurIPS2023-AI4Science Poster_

### Official Review · Reviewer_sgqg · 2023-10-19
**Review of Predicting the Initial Conditions of the Universe using a Deterministic Neural Network**

**Rating:** 7
**Confidence:** 2

**Review:**

The authors propose to use a neural network to deterministically learn the (traditionally one-to-many) output->initial conditions mapping of a cosmological simulation. The authors show that the initial conditions of many classes of simulations can be recovered to a surprising degree of accuracy. Despite the significant physics background involved (and this reviewer's unfamiliarity with the field), the authors do a good job of distilling the central ideas of the paper to a broader ML audience.

---

### Meta-Review · Area_Chair_EDbd · 2023-10-27

**Recommendation:** Accept (Poster)
**Confidence:** 4

**Metareview:**

Authors present an interesting application of neural networks, which they claim as pioneering work, for mapping outputs to initial conditions for an N-body cosmological simulation. The paper is cleary written and organised, however, authors should consider to add baselines that are generally used to solve this inverse problem.
Keeping in view the novel perspectives brought by the paper to solve an important problem of reverse N-body simulations, I recommend acceptance.